# Do Homicide Perpetrators Have Higher Rates of Delayed-Suicide Than the Other Offenders? Data from a Sample of the Inmate Population in Italy

**DOI:** 10.3390/ijerph192416991

**Published:** 2022-12-17

**Authors:** Silvia Raddi, Francesca Baralla, Alberto D’Argenio, Simona Traverso, Marco Sarchiapone, Marco Marchetti

**Affiliations:** 1Department of Health Sciences, University of Florence, 50134 Florence, Italy; 2Department of Humanities, Social Sciences and Education—SUSeF, University of Molise, 86100 Campobasso, Italy; 3Department of Neuroscience, University of Rome Tor Vergata, 00133 Rome, Italy; 4Department of Medical Science, Chirurgical and Neuroscience, University of Siena, 53100 Siena, Italy; 5Department of Medicine and Health Sciences, University of Molise, 86100 Campobasso, Italy

**Keywords:** inmate’s suicide, homicide–suicide, homicide-delayed suicide, aggressive and violent behavior, suicide prevention

## Abstract

Homicide-suicide can be defined as homicide followed by the suicide of the perpetrator shortly afterward. In the so-called “homicide-delayed suicide”, homicide and suicide occur but within a wide and not strictly defined timeframe. This study analyzes data concerning the suicide of 667 inmates in Italy between 2002 and 2015, considering homicide perpetrators compared to all offenders. The analyses revealed that inmates who had committed homicide were more likely to commit suicide (71% versus 45%; χ2 = 10.952, *p* = 0.001) and the odds of suicide increase concerning 1.58 times among homicide perpetrators. The time-to-suicide interval after homicide ranges between 0 to 9.125 days (mean = 1.687,9; SD = 2.303,1). Moreover, the intimate-homicide offenders who committed suicide had a significantly shorter survival time after the offense than did the other non-intimate offenders who died by suicide (*t* test, *t* = −3.56, df = 90, *p* = 0.001). The link between homicide and higher suicide risk in homicide perpetrators should be highlighted because of all the homicide offenders passing through the criminal justice system. Superior knowledge about the path of homicide-delayed suicide will be of particular use to professionals in evaluating and treating homicide inmates.

## 1. Introduction

Prison inmates are recognized as a population with a high burden of disease from a wide range of physical and mental health problems e.g., [1]. Studies underlined that mental disorder and substance abuse, but also criminal behavior, are independently associated with an increased risk of premature death [2,3,4], specifically suicide [5].

Previous studies have examined and documented the link between criminal behavior and increased risk of premature death, e.g., [6,7]. Investigations of suicide among a community of violent offenders have also suggested that this population may be even more vulnerable to death than prisoners that are non-violent offenders [8]. To date, also knowing that homicide offenders have a considerably increased mortality rate than the general population, few studies have addressed this question, e.g., [9,10]. Indeed, few studies have considered that homicide offenders have a notably increased risk of premature and violent death, especially by suicide e.g., [4]. Specifically, it has been demonstrated that offenders have a considerably increased mortality rate compared to the general population, especially by suicides and accidents [11,12]. An evaluation of prison suicide’s rate among homicide offenders may reveal that homicide–suicide could be more common than declared and this association needs clarification and updating [13].

Homicide–suicide was defined as the homicide of one or several victims, followed by the suicide of the perpetrator shortly afterward [14,15,16,17]. Homicide–suicide represents a sub-category of homicide and a serious form of interpersonal violence [18] with aspects that distinguish them from both homicides and suicides [15,18,19]; moreover, it is widely considered to have more in common with suicide [20,21]. Several definitions of the homicide–suicide act have been proposed since the 1950s, e.g., [22]. One important issue is that the suicide preceded by homicide does not result as a homicide–suicide in a criminal charge [1,18,19], because they were not always recorded as related. Accordingly, to Liem [15], the theoretical understanding of homicide–suicides is limited because homicide and suicide are perceived as fundamentally different in nature.

Specifically, as Prabha Unnithan and collaborators underlined [23], the research on homicide has neglected suicide and the research on suicide has neglected homicide. Even less has be conducted to describe the intersection of the two behaviors, and also in the case of homicide followed by suicide [24]. The assumption of a common matrix existing between homicide and suicide dates back almost a century ago: in 1886, Morselli [25], an Italian psychiatrist and anthropologist, formulated the so-called “water stream theory”, in which affirmed that suicide and homicide are “two streams of water drawing from the same source”. Coid [26] described the trend of the phenomenon and proposed that the higher the homicide rate in a population, the lower the rate of homicide–suicide. In line with the existing data on lethal violence, e.g., [23,24], the term “delayed” suggest a connection between other-directed and self-directed violence, considering an extended timeframe between homicide and suicide. To date, there are broad variations in operational criteria for homicide followed by suicide, e.g., [15]. Most findings underlined that perpetrators use 24 h [19,27], or a few days [28], as a time interval between homicide and suicide. Some authors, instead, include suicide-events within one week after homicide [29]; others, used no time limit at all [30]. Suicide may occur simultaneously or even before in the suicide–homicide [31], in suicide by pilot [32,33], and in suicidal attacks [34,35,36].

Ever since West [20], few studies mapped the incidence and prevalence of homicide–suicide in persons who kill within relationships, compared to those who kill a stranger, and tested the hypothesis that the former are more likely to commit suicide than the latter. Further ties between homicide and suicide have been reported from the psychopathological perspective, e.g., [37]. On the other hand, the literature concerning the causes of suicide is extremely extensive; it is not the purpose of this work to discuss these contributions.

In this work, we have analyzed a form of homicide–suicide that can be termed “delayed”, whereby homicide and suicide occur, but within a wide and not strictly defined timeframe and during detention among homicide offenders. Our research presents a record and analysis of all suicides committed between 2002 and 2015 in Italy among the inmates’ population. The average suicide mortality rate in Italy is 8.15 per 10,000 inmates in the total prison population, with differences in gender (8.12/10,000 for men and 9.14/10,000 for women), and nationality (8.34/10,000 for Italians against 7.76/10,000 for foreigners). We aimed to examine whether individuals who have committed homicide have a higher suicide risk rate than the general correctional population. Moreover, we aimed to evaluate the timeframe that has elapsed between the homicide and the suicide and the difference related to the type of relationship (Intimate vs. Non-Intimate) between the offender and the victim(s). In this contribution, we hypothesized that the suicidal rate may be higher among homicide offenders and that the time-interval between homicide and suicide may vary according to the type of relationship with the victim (Intimate vs. Non-Intimate).

## 2. Materials and Methods

For our research, we consulted the list of dead inmates between January 2002 and December 2015, supplied by an association that deals with the health and rights of prisoners and provides data on its website (www.ristretti.org; accessed on 12 June 2016) related to the inmates’ causes of death. The data comprised the name, last name, age, date of birth, prison states and cause of death. Further information related to the inmates’ suicides (crime committed and its characteristics, as well as legal position) was collected from the same website and also from daily national and local papers that report the news.

To improve the accuracy of the data, the reliability of the information was checked through several alternative newspaper reports and websites. Thus, much information in the study was not available, and the absence of an indication of a characteristic was coded as “unknown”.

For each suicide, the data collected included: the name of the person who committed suicide, his/her nationality and the date of death, and cross-referenced the following data: nationality, region or country of origin, crime committed, legal position, suicide modality and sentence length. For convicted subjects, we also evaluate the remaining sentence. Information about the victim was collected for homicide: the victim’s acquaintance and the type of the relationship with the author of homicide.

In the present research, the term “convicted” was exclusively related to inmates who have been proven guilty; the term “awaiting sentencing” refers to inmates in all other stages of judgment without distinction.

Our data do not contain a psychiatric and/or psychological evaluation, a clinical and physical condition and observation, or alcohol or drug-related disorder diagnoses, as well as the presence of previous suicide attempt/s. All offenders interned in judicial psychiatric hospitals were excluded from this count.

The total correctional population data are derived from the Italian Department of Prisons Authorities (D.A.P.; Ministry of Justice, web site: www.giustizia.it/giustizia/it/mg_12_3.wp; accessed on 20 June 2016), and by the National Institute of Statistics (ISTAT; web site: http://www.istat.it/it/; accessed on 24 June 2016). All the correctional population demographics’ data used (e.g., number of convicted for each year; the number of convicted related to the type of crime for each year) were achieved as an average between values referred by ISTAT; therefore, they must only be considered as estimates. The Italian yearly correctional population considered in our research from 2002 to 2015 consisted of 58.200 inmates, 95% males and 5% females. Foreigners are approximately 34%. As to the composition by age groups, the population was calculated as follows: 1.1% with 18–20 years of age, 6.9% with 21–24, 13.2% with 25–29, 15.8% with 30–34, 15.95% with 35–39, 14.8% with 40–44, 11.4% with 45–49, 13.55% with 50–59, 5.3% with 60–69, and 2% with 70 years and over. Approximately 70% of inmates are in prison for crimes related to property. The rate of convicted homicides or attempted homicides is 14% among the Italian convicted population and 8.5% among foreigners. Among the total sample, 50% of detainees received a final verdict.

SPSS^®^ version 24 statistical software for Windows was used to analyze the data related to the rates and patterns of suicide among prisoners. Distributions of individual items were assessed, including missing data, in the adult prison. A descriptive analysis of the sample was carried out. Categorical variables are presented as frequencies (%) and absolute numbers (N). To investigate the association and the differences between the factors considered, we used Chi-square tests, the Odds ratio and the *t* Test, as appropriate. For all analyses, statistical significance was predetermined at α < 0.05.

The analyses were performed considering the group of homicide offenders as reference subgroups. We included attempted homicides following the homicide–suicide classification by Traverso [38]. The descriptive variables used were the suicide event itself, with the indication of other available data concerning: age, gender, nationality, sentence length, time spent in prison, and the type of crime committed. Among the group of homicide perpetrators, the data also included the timeframe that has elapsed between the homicide and the suicide, as well as the type of relationship between the offender and the victim(s). Specifically, victims were classified as known (VK) or unknown (VU). Whenever multiple victims were present, some of them known and others not, the homicide was classified separately. Regarding the relationship between the victim and the homicide, it was classified as Intimate (VI) or Non-Intimate (VN-I). The victim(s) were considered as intimate if they included a partner (current or former spouse, girlfriend or boyfriend) or a (biological, step or foster) child, parent, stepparent, sibling, or another family member. Homicides involving multiple victims falling within both categories were considered separately.

To assess the association between the number of suicide and the type of crime committed (“Homicide” vs. “Other Crime”), considering the general convicted population, we used two dichotomous (0 or 1) outcome variables: the primary related to suicidal behavior in prison (“Suicide”/“No Suicide”), and the secondary related to the type of crime committed “Homicide”/“Other Crime”. We used the estimated mean of all the convicted population with or without suicide in the convicted homicide offenders compared to all offenders during the reference period between 2010 and 2015 (for which the statistical results are collected and to which the data refer) to perform Pearson’s Chi-square test with an Odds Ratio estimation. We considered also the timeframe that has elapsed between the homicide and the suicide among homicide offenders.

## 3. Results

Inmates who died in prison for all causes from 2002 to 2015 were 1.296, with an average of 93 deaths per year. Available data related to the declared causes of death among inmates reveal that 667 (51%) committed suicide, 339 (26%) died of a sickness, 32 (3%) died of a drug overdose, and 14 (1%) were assassinated during imprisonment. For 244 (19%) inmates, the cause of death is unknown. For the 667 suicide inmates, hanging was the most widely used suicide modality (80.9%). The other suicide modalities, were suffocation (14.7%), poisoning (1.6%), precipitation (0.9%), exsanguination (0.7%), shooting (0.6%), stabbing (0.1%), and burning (0.1%).

### 3.1. Characteristics of Inmate Suicides

The 667 inmates’ suicides are 95% male and 5% female.

As regards to nationality, 67% of suicides are Italian and 33% foreigners. Among the 33% of foreigner inmates, they are mainly African (48%) or Eastern European (34%), followed by Latin American (8%), Western European (6%), and Asian (4%), and one suicide came from North America. A total of 60% of Italians come from southern Italy, followed by northern and central Italy. In fifty-one cases (7,6%), the place of birth was unknown.

As to age, overall, male suicides mainly fall within the 18–29-year age group (26.3%), while female suicides fall within the 40–44-years of age group (31.25%).

### 3.2. The Crime Committed

About 29% of inmate suicides were in prison for property crimes (Table 1). Homicide and attempted homicide hold second place and represent the charge for 25% of suicides (138 cases), followed by drug-related crimes (20%), organized crimes (9%), sexual crimes and stalking (8%), and other crimes against life (3%), as well as immigration-related crimes (3%).

Our sample included 28 subjects who had committed one or more than one attempted homicides, and 116 who had committed one or more than one homicide.

### 3.3. Characteristics of Suicide among Homicide Offenders

The 138 homicide perpetrators’ suicides are 95% male and 5% female.

According to nationality, 72% are Italian and 28% are foreign. This proportion varies if we stratify it by gender. The rate for Italian females goes up to 83%. Foreign suicides convicted for homicide mainly come from Eastern Europe (50%). About 60% of Italians come from southern Italy, followed by northern and central Italy.

As to age, when considering the percentage of prisoners for homicide, we may note that they are older than other suicides. A total of 36.11% of inmates’ suicides were an over-60 age group, 32.91% from 50 to 59, and 29.09% from 45 to 49 years of age. Moreover, compared to the other crime (mean = 37.32; SD = 11.19), the age among homicide perpetrators (mean = 42.19; SD = 12.19) does not differ significantly (*t* test, *t* = −4.48, df = 584, *p* = 0.164).

### 3.4. Mortality Rates

The average mortality rate for suicide is 8.15 inmates out of 10,000 in the total correctional population (Table 2).

For males, the rate overlaps that of the total population, with an average of 8.12 suicides every 10,000 inmates. For females, the rate was higher, with an average of 9.14 cases every 10,000 female inmates. Stratifying by nationality, the average rate was 8.34 out of 10,000, higher than that of foreigners, accounting for 7.76 out of 10,000. Stratifying by crime committed, the mortality rate for suicide is approximately double in the under-population convicted for homicide and attempted homicide.

Specifically, using the estimate mean of all convicted population with or without suicide (“Suicide”/“No Suicide”) among the convicted homicide offenders compared to all offenders (“Homicide” vs. “Other Crime”), we performed a Pearson’s Chi-square test with an Odds Ratio estimation. A total of 71% (N = 65/9.183) of deaths among homicide offenders were attributable to suicide, compared to 45% (N = 240/53.582) of deaths among other offenders. Prisoners who had committed homicide were more likely to commit suicide than the general correctional population (χ2 = 10.952, *p* = 0.001). The analyses also revealed that the odds of suicide increase concerning 1.58 times among homicide perpetrators

### 3.5. Imprisonment Length

Data related to the legal position underlined that 51% of suicides were awaiting sentencing, whereas 49% had already received the final sentence; this proportion reversed in the group of homicide offenders.

The time elapsing between imprisonment and suicide for 147 suicides is unknown, and seven suicides’ cases were homicide offenders. A total of 27% of suicides had been in prison for a while between two to six months, and 18% between 12 to 24 months. This breakdown episode in suicide behavior was observed among 144 suicides convicted for homicide. However, it was noticed that the proportion of suicide prisoners among persons detained in prison for homicide increases as the imprisonment length was extended (Figure 1).

The remaining sentence was assessed for 277 prisoners. Sixty-nine of them were convicted for homicide. In 60 cases (20%), no information was found regarding the remaining sentence. Seven of them only had been convicted of homicide/attempted homicide.

The majority of suicides had occurred when less than two years were left before prison release (32%). Vice versa, homicide offenders committed suicide quite some time before their release (23.5%; ten years before release). If we observe the rate of suicides concerning the remaining sentence in prison, we note that as the remaining sentence goes up, the proportion of homicide offenders increases (Figure 2).

### 3.6. Victims’ Characteristics

A total of 85 homicides and 22 attempted homicides (74.3%) involved one victim, 13 homicides and 2 attempted homicides (10.4%) with two victims, 2 homicides (1.4%) with three victims, and 9 homicides and 2 attempted homicides (4.8%) with four or more victims. No information was found about the number of victims of seven murders and two attempted murders.

Regarding the gender of the victims, we observed that females killed males in a double number of cases. Males murdered males and females almost without distinction. In so far as the 28 multiple murders, gender was unknown in four of them; eleven had at least one adult woman among their victims, four only had adult women as their victims, seven had only male adults, one had one child only (two boys), and one had both adult women, children, and male adults. The relationship between the offender and the victim is unknown in 14 cases. In the majority of homicides (66% in the case of female perpetrators and 51% among male perpetrators) the perpetrators knew their victim. Regarding the relationship with the victim, in our sample, 40.6% had an intimate relationship (VI). In fifteen cases, the relationship between the victim and the offender was unknown. The relationship with the victims was more frequently non-intimate (59.4%, considering the offenders of both genders). The relationship was mainly non-genetic: 49% of the victims were partners or spouses, 16% were mothers; more rarely, 6% and 4% were children and brothers or sisters, respectively.

### 3.7. The Timeframe between Homicide and Suicide among Homicide Offenders

Data related to the timeframe that has elapsed between the homicide and the suicide among homicide offenders show that it ranges between 0 to 9.125 days (mean = 1.687,9; SD = 2.303,1). Considering the relationship with the victim of homicide, as Intimate (mean = 693.03; SD = 1105.4) or Non-Intimate (mean = 2221.74; SD = 2480.7), Intimate-Homicide Offenders who committed suicide show a significantly shorter survival time after the offense than did the other Non-Intimate Offenders (Figure 3; *t* test, *t* = −3.56, *df* = 90, *p* < 0.0001).

## 4. Discussion

Suicide is one of the leading causes of death in prison and represents an international problem [13]. Studies that faced the issue of suicide in prison have indirectly confirmed the presence of a higher suicide rate among prisoners for homicide than other inmates [4,39,40,41]. The suicide-risk data among the prisoners convicted revealed that risk is higher when inmates are alone in their cells, e.g., [42], or located in a disciplinary cell, e.g., [43]. In addition, those convicted for violent and sexual crime behavior, e.g., [44], and those serving long sentences, e.g., [45], present generally a much higher risk. Our research stems from the hypothesis to verify the association between homicide and suicide’ risk, considering suicide episodes that occur during detention among prisoners detained for homicide/attempted homicide and compared to other types of crime.

Whereas the homicide–suicide phenomenon is mostly studied, e.g., [4,10,17,46,47], few authors have considered the homicide-delayed suicide. The evaluation of the “delayed-suicide” in homicide offenders may reveal that the rates of homicide-suicide are more common than declared. In the so-called “homicide-delayed suicide”, homicide and suicide are subsequent but may occur during a not strictly defined time interval. Specifically, in line with existing data on lethal violence [24], the term “delayed” suggests a connection between other-directed and self-directed lethal violence, even when considering an extended timeframe between homicide and suicide.

Several recent assumptions underpin the organic or neurobiological basis of violence related to the self and/or to others. Bourgeois [48] found a relationship between homicide and/or suicide and low levels of serotonin. Gilligan [49] postulated the role of testosterone levels in suicide. Raine [50] argued that the environment is not the only element playing a role in the origin of violence, but so is the individual’s biological predisposition. Research on twins [51,52] has demonstrated that the heritability rate between homozygote twins for a violent character is between 0.4 and 0.5, approximately, i.e., 40–50% of antisocial behavior is genetically determined. These data do not depend on the environment, since it is statistically relevant even if twins are separated at birth [53]. Empirical data on homicide and homicide offenders underlined that criminal behavior, mental disorder and substance abuse are independently related to an increased risk of premature death, in particular by suicide [3,5,54]. The evaluation of suicides’ risk in prison related to homicide behavior is a neglected research area, because it is difficult to assess the risk factors of suicide relative to homicide. Furthermore, suicide preceded by homicide does not result as homicide–suicide in a criminal charge, because they take place in different times and were not recorded as related.

Our study highlighted the link between homicide and higher suicide risk in inmates’ population. Several studies have focused on deaths occurring in prison and have underlined that prisoners are more likely to die prematurely, particularly of suicide, e.g., [6]. Existing studies on homicide–suicide have found that people who commit homicide are especially prone to suicide, and present a high death risk, e.g., [55]. More recently, in line with our findings, a systematic review and meta-analysis [56] synthesized the risk factors for suicide in prison and included, with respect to criminological factors associated with suicide risk, remand status and offence type, particularly homicide. Our findings demonstrated that prisoners who committed homicide were more likely to commit suicide than the general correctional population. Specifically, findings emphasize that the elapsed time in prison was significantly shorter among Intimate homicide inmates’ offenders compared to the Non-Intimate ones. Our data seem to confirm a particular link between homicide and suicide in inmate’s homicide perpetrators.

As previous research underlined [57], homicide–suicide was usually an event that was hard to prevent, but in line with this data, suicide prevention programs in prison must take into consideration the high suicide-risk associated with the presence of previous homicide behavior in the convicted population, in particular among Intimate homicide inmates’ offenders. Data emerging from our research highlights how Intimate homicide inmates’ offenders tend to commit suicide within the first two years of detention, if compared to Non-Intimate ones, and this is a key element that could allow the implementation of specific intervention measures.

The suicide rate in Italian prisons is much higher than that of the general Italian population. Italy is one of the countries with the lowest suicidal rates among the general population worldwide, with a standardized 4.7% rate per 100,000 inhabitants (7.6 for men and 1.9 for women; WHO, 2014). The rate among inmates is 17 times higher on average than the general population: for males 81.52 for every 100,000 prisoners (10.7 times higher), and in women, it is 91.39 for every 100,000 women inmates (over 40 times higher). In our sample, offenders are generally Italian or European middle-aged white men, whereas the victims are generally adult women, wives or partners of the murderer. Hanging is the most frequently used suicidal modality in prison (higher than 80% of cases). These data are not surprising given the shortage of self-endangering means available in prison, and research findings support this trend e.g., [4,15]. The suicide rate of Italian prisoners (83.43 per 100,000) shows no significant differences with the rate of foreign prisoners (77.57 per 100,000) (ratio = 1.07). Homicide/attempted homicide offenders represented 25% of the sample. The given correctional homicides’ population represents slightly more than 11% of the total number of prisoners (14% among Italians and 8.5% among foreigners). Distribution by nationality does not show significant differences, since it overlaps the correctional population rate between Italians and foreigners (76% and 34%). Foreign homicides mainly originate from Eastern Europe, in line with the homicide rate in this region [41].

In our data, we found that the number of suicides among homicide inmates’ increases, as the imprisonment length was extended. Concerning the imprisonment length, which was considered as a risk factor, data demonstrate that the most critical times are those immediately after entering prison and close to court hearings. This research had not considered the period when hearings were held, and the imprisonment length has been roughly calculated. In the total suicide population, we noticed two peaks between two to six months and one to two years, respectively, which might correspond to a time described in the research findings as periods of increased risk. However, looking at the rate of homicides, suicides make it clear that the proportion increases as the duration of time spent in prison. On the one hand, this can be explained by longer sentences for the type of crime; on the other hand, it seems to exclude the relevance of broadly studied risk factors such as the entrance into prison and the time of the verdict. Regarding the remaining sentence as a risk factor, it should be noted that whereas the majority of suicides had a remaining sentence of fewer than two years, the proportion of homicides’ suicides increases as the remaining sentence goes up.

Let us consider the fact that in Italy in 2015 prisoners without a final judgment and awaiting the sentencing state were 34% among the total inmates’ population. We found that suicide occurs in 51% of cases among those convicted of homicide and in awaiting sentencing conditions, which identifies another risk factor.

Our study has several limitations. First, the 667 suicide included all the deaths resulting from suicidal acts committed in prison. The data we used were obtained from files not collected explicitly for our study and aggregated through several institutional sources, online search engines, and national and local papers. Second, we use data recorded in the public administrative database, and the lack of complete data regarding all cases was linked to the absence of related-official data. Third, health status, hospitalizations, and alcohol or drug addiction information among inmates is not available. In addition, records of stay in the punishment cell and/or other data concerning prisoner management (e.g., visits from relatives) is not available. Fourth, events occurring since incarceration or related to the previous incarceration, such as experiences of self-injury, attempted suicide, and/or suicidal behavior, are not available.

The major limitation of our study is that the prisoners could not be classified according to mental-ill health, as well as drug dependence. Specifically, the indication of a person at risk for drug-related deaths was unknown. Furthermore, homicides admitted to judicial psychiatric hospitals were excluded from our study.

The presence and the relevance of mental illness was not considered in our study due to the absence of data. For this reason, among our sample, an unverifiable relevance of psychopathological factors may be assumed. The correlation between homicide and mental illness exists and has been reported, e.g., [58,59,60].

Suicide is a leading cause of mortality in prisons worldwide [58,59], and suicide rates among inmates are at least three times higher for men and nine times higher for women compared to the general population [13]. Zhong and collaborators [54], in a recent meta-analysis, considered the result of 77 research data assessing the individual risk factors of suicide in prison, comparing prisoners who died by suicide with people of similar age and sex who are living in the community. The main risk factors identified were lethal suicide attempt(s) [60]. People who attempt or commit suicide meet also the criteria for at least one psychiatric disorder, roughly from 94 to 98%, e.g., [61,62]. Major depression and bipolar disorder are associated, respectively, with a 5.1 and 4.6 relative risk of suicide deaths between adult men and women [63], followed by schizophrenia, schizoaffective disorder [64], and substance abuse [65]. A recent study [66] pointed out that depression seems to be the most common disorder specifically in homicide–suicide behavior. The psychiatric symptoms are common among homicide offenders, but the reported share of mental disorder varies between different studies, e.g., [4]. Mental disorders, mainly linked to homicide, seem to be substantially the same ones existing as risk factors in suicide [67]. In a recent review [68], the authors considered 36 risk factors among suicide attempts in prison and found a strong association for suicidal ideation, previous self-harm and markers of psychiatric morbidity, as well as specific prison-related risk factors, which included solitary confinement, victimization and poor social support while incarcerated.

In line with the previous contribution [24], the thesis that the closer the ties between the offender and the victim in a homicide, the greater the likelihood of suicide, our data also highlight that the Intimate-homicide offenders show a significantly shorter survival time after the offense than did the other Non-Intimate ones. The basic proposition, that the greater the strength of the relationship, the higher the risk of homicide, has been extensively illustrated in criminology theory since the Henry and Short works’ [69]. The Authors do not provide any systematic analysis of the phenomenon of homicide followed by suicide and, however, offer a hypothesis for why some homicides might be followed by suicide (p. 117). In our study, although we could not consider the reasons (due to the absence of data), we highlighted an important association between the type of relationship between victim and offender (Intimate vs. Non-Intimate) and the time elapsed between homicide and suicide in prison, demonstrating a significantly shorter survival time after the offense among Intimate offenders.

The data suggest that prisons’ suicide are likely to be the result of a complex interaction of different factors. Based on our finding, future research should investigate the associations between offence categories and prison’ suicide rates. Specifically, our data, in line with the previous work, e.g., [24], underlined the intersection of two behaviors: homicide followed by suicide, in the named “delayed” form. We have contemplated homicide delayed-suicide by taking into account an extended timeframe between homicide and suicide and considering the suicide related to homicide inmates. The lack of work related to this criminal behavior is associated with the rarity of the event and the difficulty detecting a homicide followed by a suicide, even when the suicide occurs in prison and during detention. Suicide preceded by homicide does not result as homicide–suicide in a criminal charge because they were not recorded as related, especially if they occurred within a wide and not strictly defined timeframe. A large sample of homicide offenders suicides’ need to be evaluated during imprisonment in order to identify an adequate number of homicide–suicide. Underlying these factors could be informative and the evaluation of their interaction might also provide some explanation.

A comprehensive knowledge of what factors may increase the risk of suicide among inmates can inform prevention efforts to reduce mortality in this high-risk population. Preventive interventions should target these risk factors associated with suicide in prison and might improve the assessment of risk, risk stratification, as well as resource allocation in prison services. In Italy, all inmates received adequate health assistance [70], but clinical records do not usually report the committed crime. Our data suggest that for homicide offenders, the crime itself represents a specific risk factor; thus, it should be taken into consideration in a suicide risk assessment.

## 5. Conclusions

Taking into consideration the data of all the suicides committed in Italy between 2002 and 2015 among prisoners, it emerged that the individuals who have committed homicide have a higher suicide rate risk compared to the general correctional population. Moreover, we found that Intimate homicide offenders who committed suicide in prison show a significantly shorter elapsed time after the offense than did the other Non-Intimate offenders. We have contemplated homicide–suicide in a “delayed” form, taking into account an extended timeframe between homicide and suicide, and considering the suicide related to homicide inmates’. The large sample of suicide among homicide offenders needs to be evaluated during imprisonment in order to identify an adequate number of homicide–suicide. Considering the link between homicide and higher suicide risk in homicide perpetrators, the homicide delayed-suicide should be highlighted because almost all homicide offenders pass through the criminal justice system. The superior knowledge about the path of homicide-delayed suicide will also be of particular use to professionals in improving a specific assessment of risk and resource allocation in prison services.

## Figures and Tables

**Figure 1 ijerph-19-16991-f001:**
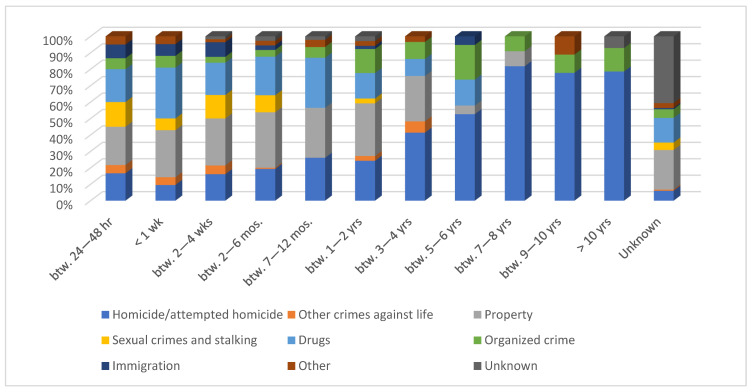
Type of crime and suicide percentage, considering the imprisonment length.

**Figure 2 ijerph-19-16991-f002:**
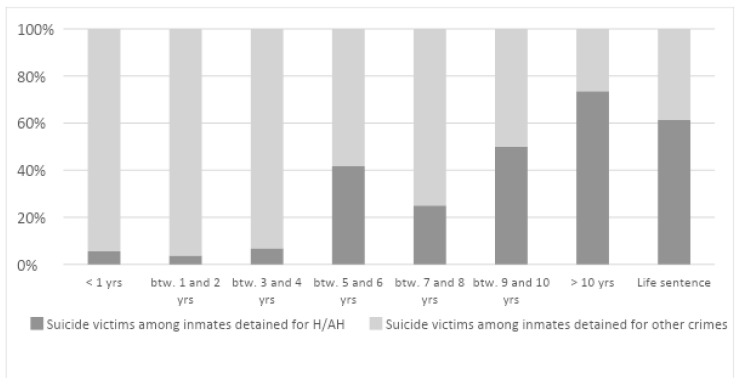
Suicide percentage considering the time sentence in inmates detained for other crimes or for Homicide/attempted homicide.

**Figure 3 ijerph-19-16991-f003:**
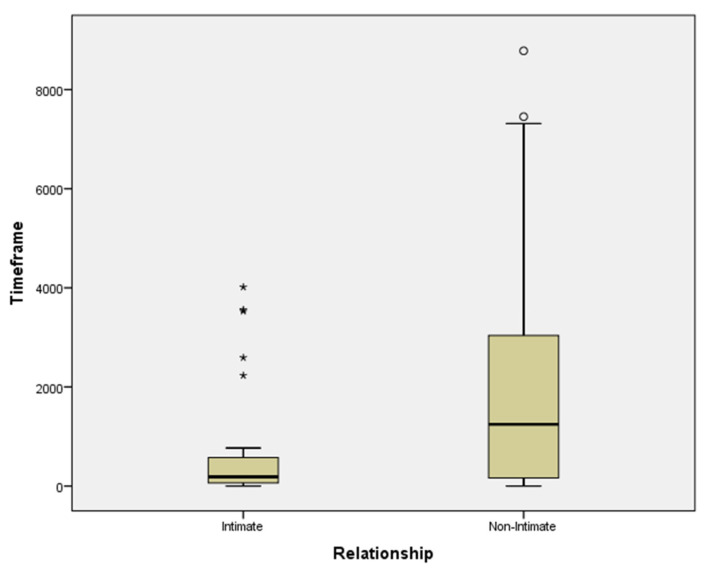
Homicide offenders and timeframe with suicide considering the relationship with the victims (Intimate/Non-Intimate). * indicated the inmates’ suicide cases distribution considering Intimate relationship ° indicated the inmates’ suicide cases distribution considering Non- Intimate relationship.

**Table 1 ijerph-19-16991-t001:** Frequencies of type of crime committed among inmates’ suicide.

Type of Crime	Frequency %
Organized crime	9%
Crimes related to substance	20%
Sexual crime and stalking	8%
Property crimes	29%
Immigration-related crimes	3%
Homicide/attempted homicide	25%
Other crimes against life	3%
Other crimes	3%

**Table 2 ijerph-19-16991-t002:** The Suicide Mortality Rate (SMR per 10K.) among all inmates’ population in Italy and specific category (total and Homicide/attempted homicide) per year (from 2002 to 2015) considering gender and nationality.

Year	SMR in all Inmates	SMR in Male Inmates	SMR in Female Inmates	SMR in Italian Inmates	SMR in Italian H/AH Inmates	SMR in Foreign Inmates	SMR in Foreign H/AH Inmates
2002	6.28	6.19	8.18	6.92	12.81	4.79	7.04
2003	7.40	7.37	7.97	7.77	11.10	6.54	14.00
2004	8.99	8.86	15.50	8.93	16.88	9.12	13.42
2005	9.44	9.19	14.54	10.93	21.79	6.35	18.67
2006	8.07	8.27	4.06	8.73	12.06	6.20	0.00
2007	9.34	9.29	10.40	7.80	15.20	12.11	37.50
2008	5.93	5.43	16.88	7.98	16.90	2.98	5.84
2009	10.45	10.42	11.17	10.23	18.27	10.83	5.10
2010	9.85	10.30	0.00	11.81	13.50	6.49	14.31
2011	8.45	8.68	3.47	7.68	11.64	9.81	14.43
2012	8.74	8.50	14.23	8.70	18.47	8.81	14.80
2013	6.95	7.26	0.00	4.54	6.82	11.37	10.29
2014	7.23	7.20	7.95	8.31	16.69	5.11	18.03
2015	7.00	6.71	13.59	6.48	18.10	8.08	20.36

## Data Availability

The datasets generated for this study are available on request to the corresponding author.

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
