# Peer review of "Do Homicide Perpetrators Have Higher Rates of Delayed-Suicide Than the Other Offenders? Data from a Sample of the Inmate Population in Italy"

_ijerph, 2022, doi:10.3390/ijerph192416991_

Round 1

Reviewer 1 Report

The topic is interesting, by the way I can't understand if the paper is a commentary about the two studies performed by Centro Due Palazzi e Associazione Granelli di Senape, since the authors didn't t provide a link to check it.

About other concerns I suggest to clarify the materials and method used and if it is a commentary to highlight it; introduction section should be shorter and more focused on the aim and scope of the paper.

Author Response

Dear Reviewer,

thanks for your attention to our work and for your suggestions.

Point 1: The topic is interesting, by the way I can't understand if the paper is a commentary about the two studies performed by Centro Due Palazzi e Associazione Granelli di Senape, since the authors didn't t provide a link to check it.

Response 1: We specified in the section related to the method that we analyzed the published data supplied by an association that deal with the health and rights of prisoners and that provides data related to the inmates’ cause(s) of death on the website www.ristretti.org (see for details morti.carcere.xls (live.com). 

Point 2: About other concerns I suggest to clarify the materials and method used and if it is a commentary to highlight it; introduction section should be shorter and more focused on the aim and scope of the paper.

Response 2: We have clarified in the materials and methods section how we have collected the research data, that does not correspond to a comment on the data to which we refer. We have also made the changes in the introduction section which has also been made (hopefully) more focused on the aims and scope of the paper. 

Reviewer 2 Report

The manuscript is well-written and mostly clear.

Please check on the consistency of use of terminology, for example, "organized crime" and "Mafia crime" - is it the same thing?

Figure 1 is not clear - suppose it will be printed in color to make it clear?

Lines 231 to 235 were initially not clear to me, so maybe need to be rephrased to indicate "how many victims were included per homicide case". 

Author Response

Dear Reviewer,

thanks for your kindly attention to our work and for your suggestion.

Point 1: Please check on the consistency of use of terminology, for example, "organized crime" and "Mafia crime" - is it the same thing?

Response 1:  We have changed the terminology. In fact, we have considered data on organized crime in general, erroneously indicated in the table as "mafia crime".

Point 2: Figure 1 is not clear - suppose it will be printed in color to make it clear?

Response 2: We have edited the image and added the color version to show the data more clearly.

Point 3: Lines 231 to 235 were initially not clear to me, so maybe need to be rephrased to indicate "how many victims were included per homicide case". 

Response 3: We have considered his helpful remark and re-written the sentence relating to lines 231-235 as follows: "85 homicides and 22 attempted homicides (74.3%) involved one victim, 13 homicides and 2 attempted homicides (10.4%) two victims, 2 homicides (1.4%) three victims, 9 homicides and 2 attempted homicides (4.8%) four or more victims. No information was found about the number of victims of seven murders and two attempted murders".  We hope that it is clear now. 

Round 2

Reviewer 1 Report

I appreciate the efforts made by the authors.

I just suggest to format the references according to the guidelines of the journal and read and include in the references the following paper  https://doi.org/10.3390/healthcare9111511

Author Response

 Point 1: I appreciate the efforts made by the authors.

Response 1: We thank the reviewer for appreciating our reviewing work of the paper.

Point 2: just suggest to format the references according to the guidelines of the journal 

Response 2: Thanks to the reviewer for the tips. We have revised the references according to the guidlines of the journal.

Point 3: read and include in the references the following paper  https://doi.org/10.3390/healthcare9111511

Response 3: Thanks to the reviewer. We have read and included the reported contribution of research. 
